# ST-Trie: A Novel Indexing Scheme for Efficiently Querying Heterogeneous, Spatiotemporal IoT Data

**Hawon Chu [1], Jaeseong Kim [1], Seounghyeon Kim [1,†], Young-Kyoon Suh [1,\*]** **, Ryong Lee [2,\*], Rae-Young Jang [2] and Minwoo Park [2]**

1   School of Computer Science and Engineering, Kyungpook National University, Daegu 41566, Korea;
    hwchu@knu.ac.kr (H.C.); jskim94@knu.ac.kr (J.K.); kshy9598@knu.ac.kr (S.K.)
2   Research Data Sharing Center, Korea Institute of Science and Technology Information, Daejeon 34141, Korea;
    raezero@kisti.re.kr (R.-Y.J.); pminwoo@kisti.re.kr (M.P.)
\*   Correspondence: yksuh@knu.ac.kr (Y.-K.S.); ryonglee@kisti.re.kr (R.L.)
†   Current address: Samsung Research, Samsung Electronics, Seoul 06765, Korea; s_hyn.kim@samsung.com

**Abstract:**   Recently, various environmental data, such as microdust pollution, temperature, humidity, etc., have been continuously collected by widely deployed Internet of Things (IoT) sensors. Although these data can provide great insight into developing sustainable application services, it is challenging to rapidly retrieve such data, due to their multidimensional properties and huge growth in volume over time. Existing indexing methods for efficiently locating those data expose several problems, such as high administrative cost, spatial overhead, and slow retrieval performance. To mitigate these problems, we propose a novel indexing scheme termed ST-Trie, for efficient retrieval over spatiotemporal IoT environment data. Given IoT sensor data with latitude, longitude, and time, the proposed scheme first converts the three-dimensional attributes to one-dimensional index keys. The scheme then builds a trie-based index, consisting of internal nodes inserted by the converted keys and leaf nodes containing the keys and pointers to actual IoT data. We leverage this index to process various types of queries. In our experiments with three real-world datasets, we show that the proposed ST-Trie index outperforms existing approaches by a substantial margin regarding response time. Furthermore, we show that the query processing performance via ST-Trie also scales very well with an increasing time interval. Finally, we demonstrate that when compressed, the ST-Trie index can significantly reduce its space overhead by approximately a factor of seven.

**Keywords:**   Internet of Things sensors; environment data; spatiotemporal indexing; trie; query processing

## 1. Introduction

The recent Internet of Things (IoT) technology has advanced remarkably and has been dispatched in a wide range of aspects of our modern lives. Increasingly more sensors are being attached to devices connected to the internet, called things. Equipped with novel communication technology and rich computing power, the sensing things are being ubiquitously deployed in many urban environments [1]. Indeed, this IoT technology has realized smart environments [2], including connected vehicles [3], smart cities [4], smart homes [5], smart campuses [6], smart health [7,8], smart factories [9,10], smart farms [11,12], smart retail [13], etc., accounting for a large portion of modern society.

As many more IoT sensors can continuously collect various kinds of environmental data, such as microdust density, weather, contagion, and other chemicals, as well as traffic load, there has been a growing need to store, retrieve, and analyze a vast amount of the monitored data with spatiotemporality. For instance, if when and where the density of microdust is severe during a

day can be predicted based on the collected data; accordingly, the moving path and time of a cleaning car to remove microdust from the road can be efficiently optimized. Like this, the collected data can provide valuable insight into new kinds of sustainable application services or enhancing our daily lives for better sustainability. Therefore, it is natural to pay much attention to utilizing the accumulated data well, which would otherwise be simply discarded. To this end, quite a few platforms [14–19] have been developed to support storing and querying the immense amount of IoT spatiotemporal data.

Still, it is challenging to quickly locate the IoT sensor data intersecting user-specified spatial and temporal regions. There are several reasons for that. First, a large volume of data generated by the sensors in real time consumes huge storage space. Second, the generated IoT data show multidimensional characteristics; namely, the data include the spatial coordinates (latitude and longitude) and temporal attributes of the stationary or moving IoT sensors and the associated values observed by the sensors. Although there has been a number of techniques [20] for attaining efficient access to such spatiotemporal data, the approaches still suffer from their slow retrieval performance, space overhead (caused by a too wide index key size), and administrative overhead (from managing a distributed processing framework with several architectural components).

In this regard, we propose a novel spatiotemporal indexing scheme, termed ST-Trie, to gain efficient access to heterogeneous IoT sensor data stored in a NoSQL database (MongoDB [21]). The key ideas are to first convert three-dimensional spatial and temporal data to one-dimensional index keys expressed by 64 bit integers (while preserving their locality) and to organize the encoded integer values into a (virtual) trie (the latter idea was inspired by some previous works [22,23] that achieved efficient retrieval in flash memory page address mapping). Unlike existing schemes whose performance depends on the number of stored nodes, our trie-based indexing scheme reaches a target node in constant time, regardless of how many nodes exist in the trie. In addition, since the encoded 64 bit index keys allow for bitwise operation, our traversal is accelerated. Furthermore, the leaf nodes of our index structure are connected to each other, resulting in exceedingly fast time travel for such spatiotemporal IoT sensor data. Besides, the proposed index supports various query types, such as spatiotemporal range, *k*-nearest neighbor (*k*-NN), and top-*k* queries, in a unified way, as opposed to some existing works [24,25], which utilize an independent index suitable for each individual query type.

In our experiments, with three kinds of real datasets from all around the globe (Asia, Europe, and the USA), the proposed ST-Trie index outperforms the compared schemes by a significant margin over growing spatial and temporal ranges. Furthermore, query processing performance via our index scales very well with increasing ranges in the spatiotemporal dimension. Furthermore, when it comes to *k*-NN query processing, our index yields very flat search performance over increasing *k* values and surpasses the compared schemes by a factor of three. Despite the increase of *k*, the performance of top-*k* query processing varies little across the different datasets. Lastly, ST-Trie achieves high space efficiency when compressed, compared to uncompressed, by up to about seven times.

Our contributions can be summarized in the following.

- We propose a novel indexing scheme, termed ST-Trie, enabling fast retrieval over massive data with spatiotemporality collected from IoT sensors deployed for environment monitoring.
- We introduce a new technique of intertwining three-dimensional spatiotemporal values into one-dimensional 64 bit integer keys.
- We elaborate on constructing the proposed ST-Trie index structure with the encoded keys.
- The proposed index supports processing various query types in a unified way, such as spatiotemporal range, *k*-NN, and top-*k* queries, over the measured IoT sensor data.
- We describe the query processing algorithms for the different kinds of queries.
- We conduct asymptotic analyses on the proposed indexing and query processing.
- In our experiments with different real-world IoT sensor datasets, we demonstrate that the proposed index outperforms the compared approaches and scales well with increasingly expanding spatial and temporal ranges.

The rest of this manuscript is organized as follows. The subsequent section reviews a rich body of existing literature relevant to our work. Section 3 overviews our key ideas to build a spatiotemporal index for heterogeneous IoT sensor data. Section 4 presents the proposed index's overall structure, elaborates on our index's key operations, and conducts asymptotic analyses on the index. Section 5 proposes query processing algorithms using the constructed ST-Trie index. Section 6 shows various experimental results on the effectiveness and efficiency of our index compared to several competitors. Finally, Section 7 concludes this manuscript and suggests future directions to expand our work.

## 2. Related Work

Recently, quite a few indexes have been proposed to support efficient retrieval over large-scale spatiotemporal data. Many of the proposed indexes were based on transforming given two-dimensional longitude and latitude data into one-dimensional values. Particular emphasis was put on calculating the minimum bounding rectangle (MBR) of the specified spatial range. The proposed indexes typically took an R-tree or B-tree-like structure, such as P-tree [26], Trajtree [27], Trails-tree [28], DITIR [29], V-tree [30], and D-Toss [31].

These types of indexes tend to generate MBRs for each type of spatial data, such as point, line, grid, trajectory, etc. This means that there are many overlapping areas between MBRs, and this results in performance degradation in index management and inefficiency in query processing. Furthermore, such indexes need to be updated as the volume of data increases, which leads to high maintenance costs.

When it comes to spatial-partitioning models, S2Geometry [32] has been used for spatial data indexing. First, Li et al.'s work [33] proposed an adaptive multilevel index tree, termed MLS3, for indexing spatiotemporal data in P2P networks. The authors leveraged and extended S2 Geometry to encode spatiotemporal information in the row key of Apache Cassandra [34]. The row key was hashed to distribute spatiotemporal data in the P2P networks. Although their tree had the same spatial basis as our proposed ST-Trie, the two index structures are fundamentally different. First, their MLS3 tree was implemented in a specific NoSQL database, Cassandra. Conversely, ST-Trie is orthogonal to any NoSQL database, making ours more general-purpose. Second, the MLS3 tree was customized for P2P networks, whereas ST-Trie is applied to heterogeneous IoT sensor data with spatiotemporal attributes and is extensible to other domains, including such P2P environments. Third, their index adopts a multilevel tree structure whose depth is determined by time and spatial granaries, while our index is based on a binary trie whose height is automatically fixed along with the key size (here, 64). Lastly, the size of the row key, consisting of a partition key and a sort key in the partition, in the ML3 tree is more than 131 bits, while our index key is only 64 bits. With respect to storage efficiency, the previous work is, therefore, not appropriate for indexing immense, ever-growing spatiotemporal IoT data. On the other hand, our index key is much more lightweight and space efficient.

Guan [35] proposed the ST-Hash (spatiotemporal-hash) scheme, in which a time-space index key is generated by intertwining two-dimensional spatial information to encoded values based on the geohash. It then creates an intermediate key using the time remaining (e.g., month, day, hour, minute, and second, except the year), then adds the last year to finally generate a string-type three-dimensional space-time index key. The ST-Hash technique has a decisive disadvantage in expressing latitude and longitude information in more detail, where longer time and space index keys consume more storage space.

Qian [36] proposed GEOSOT, which is similar to ST-Trie in that it encodes and stores time and space information in an Int64-type integer. However, the minimum range of spatial information is limited to 1 arcsec since 21 bits were used to store latitude and longitude information. Therefore, it cannot store more detailed spatial information than the S2 Geometry used in this manuscript.

Binna [37] developed the HOT (height optimized trie), which is a trie-based index that combines nodes at each level to use dynamic bit numbers for fan-out. This allowed the fan-out of each node to increase and, consequently, reduced the height of the trie. However, as noted in the Introduction, there is a disadvantage where each node's fan-out overflows during the creation process,

requiring additional time to resolve it. On the other hand, the ST-Trie proposed in this manuscript ensures the insertion performance, regardless of the number of nodes being inserted.

Arseneau [25] proposed the STILT technique, using a binary PATRICIA tree to increase the fan-out of the node, as with HOT, and to reduce the height of the trie. While the ST-Trie indexing scheme proposed in this article supports the search for various types of time-space thing data with an integrated index, the previous work, STILT, uses respective indexes of different structures. These independent indexes depend on the type of query, so STILT results in significant costs for index management and maintenance, as mentioned in Section 1.

Li [24] developed the JUST system, which proposes a three-dimensional indexing technique based on Geomesa [38], extending the two-dimensional indexing technique into time and space. The JUST system indexes spatial information based on the time-space object data's time information, while ST-Trie indexes time information about spatial information. Moreover, similar to STILT, JUST systems have the disadvantage of creating indexes of different structures for different types of data on time and space objects, making them costly to manage and maintain indexes.

Table 1 provides a summary of how different or similar our work is compared to several competitor's. Our approach's biggest advantage is that it does not prefer any specific NoSQL storage; ST-Trie can collaborate with any NoSQL storage to be able to save JSON-like documents containing spatiotemporal IoT data. It also supports various types of spatial data queries in a uniform way. Furthermore, our key is fairly shorter than the competition's, so it achieves spatial efficiency in index management.

**Table 1.** Comparison with the state-of-the-art indexing schemes. ST-Hash, spatiotemporal-hash.

| Category | ST-Trie | ST-Hash [35] | XZ2T [24] | GEOSOTST-index [36] | MLS3 [33] |
|---|---|---|---|---|---|
| Spatial Encoding | S2 | Geomesa | Geomesa | Geomesa | S2 |
| Object Type | Spatiotemporal | Spatiotemporal | Spatiotemporal | Spatiotemporal | Spatiotemporal |
| Data Structure | Trie | Hash table | Table | Table | Tree |
| Index Key Size | $<= 64$ bits | $>= 80$ bits | $>= 54$ bits | 64 bits | $>= 131$ bits |
| NoSQL Dependence | No | Yes | Yes | Yes | Yes |
| Index Storage | In-memory (file system) | Cassandra | Accumulo | Cassandra | Any NoSQL database |
| Query Type | ST range, $k$-NN, Top $K$ | ST range | ST range, $k$-NN | ST range | ST range |
| Application Domain | General IoT data type | Trajectory | Trajectory | Trajectory | P2P networks |

Next, we discuss our approach to building a trie-based index structure leveraging S2 Geometry.

## 3. Our Approach: The Novel Indexing Scheme, ST-Trie

In this section, we briefly explore existing spatial encoding schemes and introduce our underlying indexing scheme. Since the chosen spatial scheme itself does not suffice for indexing spatiotemporal data from various IoT sensors, we present a novel technique for transforming the given three-dimensional spatial and temporal attributes to the one-dimensional index key. Finally, we discuss why a trie structure is suitable for our spatiotemporal IoT data indexing.

### 3.1. Leveraging an Existing Spatial Indexing Scheme: S2 Geometry

Several indexing strategies for spatial data have been proposed. For instance, geohashing [20] is one of the well-known encoding techniques for a spatial location with a short alphanumeric string. It is based on a space-filling curve algorithm, such as Z-order. GeoMesa [39], applying the geohashing algorithm, is an open-source, publicly accessible database that supports spatiotemporal data indexing in various NoSQL storage systems, such as Accumulo, Cassandra, HBase, Kafka, and Spark. Another geo-encoding technique is the World Geographic Reference System (GEOREF) [40].

The GEOREF scheme divides the entire globe into quadrangles of latitude and longitude and endows an individual quadrangle with a specific code. The unit of the geolocation area is defined by the GEOREF scheme along with the way that a concrete point is situated. The Open Location Code (OLC) [41], developed by Google, encodes a location into a Plus code, so-called because the encoded key includes a "+" character. Plus codes are derived from latitude and longitude and can often be shortened to only four or six digits. The longer the code, the smaller and more accurate the area represented. OLC is open-source and available for anyone to use. S2 Geometry [32], based on the Hilbert curve, is another spatial subdivision model, which represents all the spatial data on a three-dimensional sphere, similar to Earth, and uses a single coordinate system. Each of the spatial regions is specified with low distortion so that the entire data can resemble the true shape of Earth. Although each possesses its unique features, in this paper, we employ S2 Geometry, mainly due to its robust performance and accessibility to various libraries. Note, though, that our indexing scheme is orthogonal to any geographical encoding technique, as long as the technique can return a numeric value to represent a certain spatial area.

S2 Geometry exhibits the following characteristics. (1) Recursive: S2 Geometry defines a framework for decomposing the unit sphere into a hierarchy of S2 cells. Each S2 cell is a quadrilateral bounded by four sub-cells. The root of the hierarchy is obtained by projecting the six faces of a cube onto the unit sphere, and the S2 cells at the lower level are subdivided into four children cells, recursively. (2) One-dimensional: Each S2 cell is indexed by a 64 bit integer, rather than an alphanumeric string adopted by Geohash [20,39] based on the Z-order curve [42]. As the S2 code is not a string, it allows us to perform direct bitwise operations without any further transformation. Thus, calculating some distance and coverings with S2 is much quicker than with Geohash. Furthermore, thanks to locality preserved by the Hilbert curve, spatiotemporal queries with Hilbert keys can be processed faster than those with Z-order-based keys [43]. (3) Flexible: S2 Geometry supports flexible spatial indexing because it can approximate arbitrary regions as collections of discrete S2 cells. Based on these characteristics, we can easily and efficiently find nearby objects, measure distance, compute centroids, etc.

In the next section, we describe how one-dimensional encoding is performed on a spatiotemporal region with a specific S2 code and its associated temporal interval at which various IoT data are observed.

### 3.2. Encoding of 3D Spatiotemporal Information to the 1D Index Key via the S2 Cell ID

In our indexing, we consider extending the aforementioned S2 Geometry spatial encoding model to a novel spatiotemporal encoding model. To this end, we convert three-dimensional spatiotemporal information to a one-dimensionally encoded value to be used as our index key via an S2 cell ID. As an example, Figure 1 illustrates the transformation process. In the figure, we assume that IoT data were observed in the spatial region with a longitude value of $40.722439°$ and a latitude value of $-73.997552°$, corresponding to a district in New York, NY, USA, for a duration from 3 to 4 am on 19 September 2010.

The following is the procedure of how our 3D-to-1D encoding is performed. First, we obtain the 64 bit S2 cell ID associated with the spatial area expressed as "1000...0000". From this 64 bit numeric value, we take the first 44 bits to index a spatial region in our system. The 44 bit S2 cell ID corresponds to a Level 20 area of S2 Geometry, or about 77.32 m$^2$. Why does it correspond to S2 Level 20? When examining our IoT data, we realized that it was sufficient for a certain S2 cell at Level 20 to cover a region through which our IoT sensors were moving. Furthermore, considering the average speed of taxies, it did not matter because the values were measured at contiguous S2 cells at Level 20. Even if not contiguous, the values generally did not change dramatically between S2 cells at Level 20.

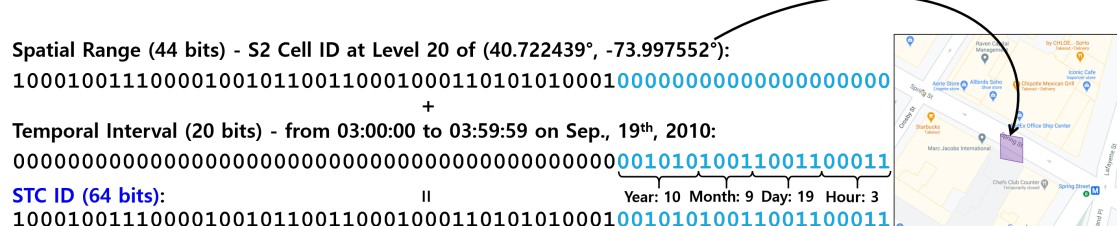

**Figure 1.** An example of transforming given 3D (spatial range and temporal interval) data to the corresponding 1D value with 64 bits.

Next, after 44 bits are used to encode the spatial part, we borrow the remaining 20 bits of the given S2 cell ID to encode temporal information. The first and second five bits are used for recording hour and day information, respectively, the following four bits for month, and the last six bits for year. A total of twenty bits is enough to index a temporal value in units of hours. In our design, the temporal part has a 64 year time window and is relative to the year 2000. However, the size of the window can be changed along with user-specified parameters.

As the final step, the encoded 64 bit 3D spatiotemporal key value is converted into an Int64-type integer. This key is used to build our trie-based index structure, which we will describe briefly.

Figure 2 illustrates the MongoDB document containing such a key value and data measured by an IoT sensor, named "SEN_01", at the spatiotemporal cell associated with the key value. This document concerns point-type spatial data and contains the averaged values of each of the five kinds of environmental things (e.g., PM 2.5, PM 10, $CO_2$, $SO_2$, and CO) measured for two seconds belonging to a certain hour (here, 3–4 am). A document like Figure 2 is stored with its spatiotemporal key value into our MongoDB database, where raw spatiotemporal thing data measured by our heterogeneous IoT sensors are collected. Nevertheless, note that our encoding scheme neither depends on a specific NoSQL storage, nor a specific relational database. The introduced scheme can be applied to spatiotemporal data with longitude and latitude values stored in any type of database.

```
{
    "_id": ObjectId("5eb2792c3a01bd678c9c5aa8")
    "Time": {
        "Start": 625,
        "End": 626
    },
    "Sensor": "SEN_01",
    "Type": "Point",
    "STC_ID": 9926594972903188067,
    "Num_of_data": 2,
    "Data_list": ["5eb2792c3a01bd678c9c5ab0",
                  "5eb2792c3a01bd678c9c5ab1"],
    "Average": {
        "PM2_5": 12.372723,
        "PM10": 13.483217,
        "CO2": 0.004135,
        "SO2": 0.005347,
        "CO": 0.287561
    }
}
```

**Figure 2.** An example of a MongoDB document containing IoT sensor data and the associated spatiotemporal index key value encoded by our scheme.

In the following section, we elaborate on why we adopt a binary trie for our indexing.

### 3.3. Why a Binary Trie

A trie (also known as a prefix tree) is a well-known popular data structure in computer science. It is an ordered tree data structure. All the descendants of a node in the trie have a common prefix of the key (typically, string) associated with that node [44]. A "bitwise" trie indicates one whose key consists of the fixed-length binary bits (0 or 1) like an integer value.

In this regard, one might ask: Why does this work take such a trie structure? Here are three major reasons for that. First, locality is the first-class citizen to be considered in our index design. In general, our IoT data reveal substantial locality in spatial and temporal dimensions. Accordingly, the index nodes pointing to IoT data that are spatiotemporally close are also likely to be neighbors in the index structure. This implies that a large portion of the index nodes' keys may be similar. Thus, the nodes end up with common prefixes in their keys, which exactly matches a trie structure.

Second, we needed an index structure in which retrieval time did not depend on how many nodes were indexed. Because IoT data are generated and stored continuously, it is not desirable to have a longer search time over the growing data. To deal effectively with the generation amount and speed of the IoT data, we solicited an index structure whose search performance is independent of the number of IoT data. The trie proved to be a perfect fit for our needs.

Besides, cache-friendliness was another important factor in our index design. If index nodes associated with the IoT data with such spatiotemporal locality could be situated together physically, we postulated that accessing the nodes together could accelerate the search performance. This made sense because other nodes not part of the very first access were more likely to be accessed soon due to the contiguity of the IoT data. Therefore, the trie structure was the best option to support our logical design. This led to our physical index design taking a cache-friendly vector (i.e., `std::vector`).

## 4. The Proposed Scheme: Search and Insert Operations of ST-Trie

In this section, we propose a novel spatiotemporal index, ST-trie, for fast IoT sensor data retrieval. We first describe the overall structure of ST-Trie and then elaborate on its search and insert operations.

### 4.1. Overall Structure

Figure 3 shows an ST-Trie structure of maximum height $h = 64$, since our index key's size, named STCID, is fixed to 64 bits. This makes the height of the trie constant. In the figure, the root node corresponds to the entire spatiotemporal range of IoT data. The upper level, from $h = 64$ to 21, indicates spatial information based on the Level 20 S2 cell ID. The lower level from, $h = 20$ to 1, represents temporal information associated with an individual S2 cell recorded in the trie. In particular, the leaf nodes at $h = 1$ have data pointers, leading to various kinds of actual IoT data (point, trajectory, polygon, etc.) collected in a specific S2 cell. This trie structure is special because the leaf nodes are connected to each other, accelerating query efficiency along with the temporal dimension. Later, we will show how the links are effective over a long time range.

In Figure 3, the arrows marked with dotted lines illustrate the nodes' jump points. If a child of a node (say, $N$), such as the left node at $h = 63$, is missing, its jump pointer points to the lowest node, such as the leftmost node at $h = 1$, belonging to $N$'s sub-trie. The jump pointer points to the leaf node with the smallest STC ID when $N$'s left child node is missing, whereas the jump pointer points to the leaf node with the largest STC ID when $N$'s right child node is missing. The route colored in blue depicts the search path via the jump pointers.

Using this index structure, we support various spatiotemporal IoT data queries in a unified way. In particular, fast bitwise operations are possible since each node represents a 64 bit long type number, enabling a more efficient search. More details about the query processing via the proposed index will be described in Section 5.

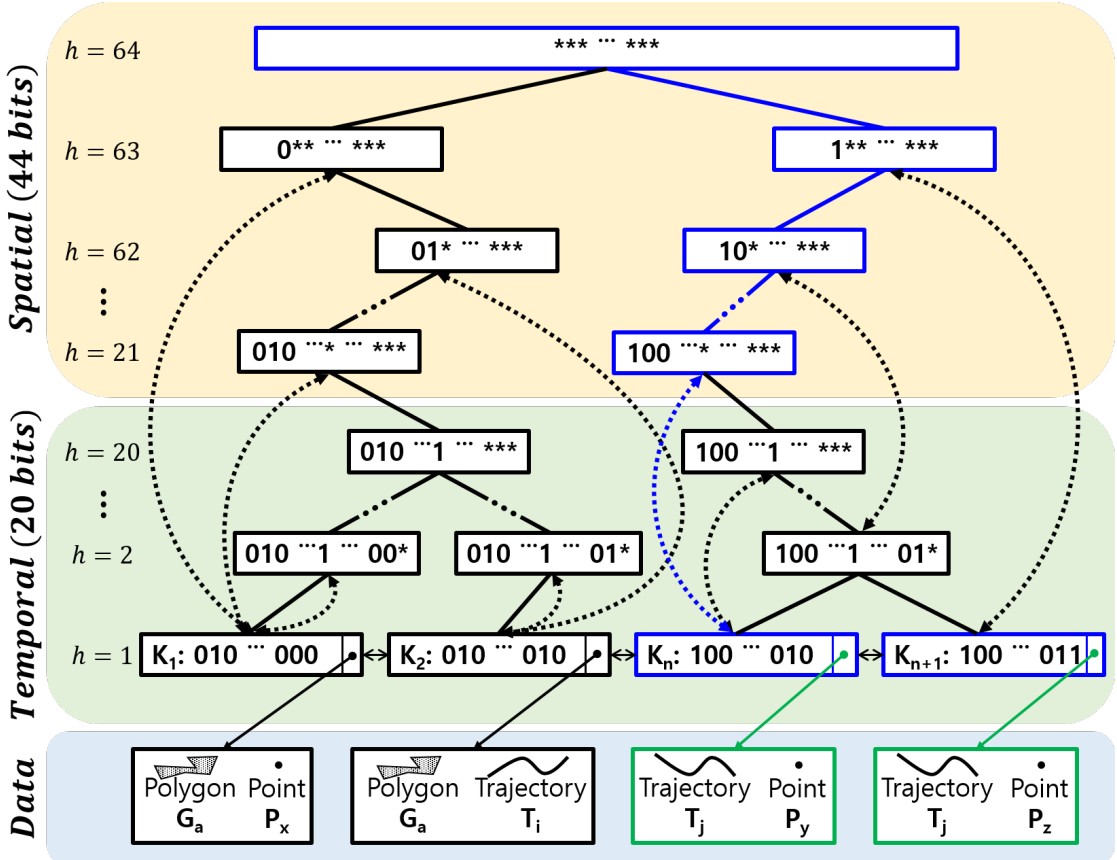

**Figure 3.** Overall structure of ST-Trie: "h" indicates the height of ST-Trie; "K" means an index key; "*" (asterisk) means the wildcard for each bit; the route colored in blue exemplifies a search path via the jump pointers; and lastly, the green arrows represent pointers to their associated actual data.

### 4.2. Searching for a Trie Index Node

Algorithm 1 specifies our node search function, named *Search*. The function takes an input search key, denoted *key*, the ST-Trie height, denoted *h*, and the ST-Trie root node, denoted *R*, and returns the found node, denoted *M*, in response to the search. First, the function performs initialization work, setting NULL to *M* and the root node *R* to the current node *N* (Lines 2–3). The search starts at the top node *N* and progresses downwards along with its child, if any, until it reaches the leaf (Lines 4–8). In the for loop, the function checks if the child, denoted *M*, of *N* exists by peeking a bit set at the *i*-th position of the input search key, *key* (Line 5). If *M == NULL*, then there is no index key associated with *key* in our index (Line 6). Otherwise, the child exists, so *N* gets updated to *M* (Line 7). The search then continues. If the search hits the bottom (Line 9), then the node matching *key* is found and thus returned (Line 10).

In the worst case, the search terminates by visiting the lowest level of the trie. Thus, the time complexity of the search algorithm is $O(h)$, where *h* is the height of the trie, a constant.

---

**Algorithm 1:** Algorithm for node search.

**input** : An input search key (*key*), the height (*h*), and the root node (*R*)
**output**: The found node (*N*)

**1 Function** Search(*key, h, R*):
**2**     $N \leftarrow NULL$;                                         `// N:  Search result`
**3**     $M \leftarrow R$;                `// Initialize a current node (M) to the root node`
**4**     **for** $i = 1; i \leq h; i + +$ **do**         `// i:  The bit position of key in walking`
**5**        $N \leftarrow M.\text{child}[\text{getBitValue}(i, key)]$;     `// Examine if the left or right child of M exists.`
**6**        **if** $N == NULL$ **then break;**          `// The corresponding child is absent.`
**7**        $M \leftarrow N$;       `// Keep walking down along with the descendant path.`
**8**     **end**
**9**     **if** $i == h$ **then** $N \leftarrow M$ ;                 `// Reached the lowest level.`
**10**     **return** $N$;                  `// Return the found node, N.`

---

### 4.3. Inserting IoT Data into ST-Trie

Algorithm 2 describes the process of inserting data into ST-Trie. It takes the same parameter *h* and *R* as Algorithm 1 and IoT data to be inserted, denoted *D*. First, generate an STC ID using the latitude, longitude, and time value of *D* as described in Section 3.2 (Line 1). Then, the algorithm performs initialization work, the same as Algorithm 1, with additional work setting *i* to one for use outside of the for statement (Lines 2–4). This involves two steps. First, check that each bit in input *K* exists in ST-Trie. In this process, *i* is storing the length of the common prefix of *key* and STC IDs in ST-Trie. If *i* and *h* are the same, then *key* already exists in ST-Trie (Lines 5–10). If not, insert *key* into ST-Trie from the *i*-th bit value to the *h*-th bit value (Lines 11–15). Then, insert *key* into the leaf node, and finally, update the links of the leaf node (Lines 16–17). When the number of data is *N*, the insertion algorithm exhibits $N * O(H)$ time complexity.

In the next section, we propose query processing algorithms via the proposed ST-Trie index.

---

**Algorithm 2:** Node insertion algorithm.

**input** : IoT data to be inserted, (*D*), the height (*h*), and the root node (*R*)
**output**: None

**1** $key \leftarrow \text{generateIdxKey}(D.lng, D.lat, D.time)$;     `// Generate an STC ID of D; Refer to Figure 1.`
**2** $i \leftarrow 1$;                            `// i:  The bit position of key in walking`
**3** $N \leftarrow NULL$;
**4** $M \leftarrow R$;                 `// Initialize a current node (M) to the root node`
**5** **for** $i \leq h; i + +$ **do**
**6**     $N \leftarrow M.\text{child}[\text{getBitValue}(i, key)]$;     `// Examine if the left or right child of M exists.`
**7**     **if** $N == NULL$ **then break;**          `// The corresponding child is absent.`
**8**     $M \leftarrow N$;       `// Keep walking down along with the descendant path.`
**9** **end**
**10** **if** $i == h$ **then return** ;                 `// Reached the lowest level.`
**11** **for** $i \leq h; i + +$ **do**
**12**     $M \leftarrow \text{createChildNode}(\text{getBitValue}(i, key), N)$ ;
**13**     $M.parent \leftarrow N$ ;
**14**     $N \leftarrow M$ ;
**15** **end**
**16** $\text{setIndexKey}(key, N)$;                 `// Set key to the index key of N.`
**17** Update the sibling pointers of the leaf node *N*.

---

## 5. Query Processing Algorithms

Our proposed ST-Trie index supports the processing of various kinds of queries over spatiotemporal IoT data. Such queries include basic spatial or spatiotemporal range, *k*-nearest neighbor, and top *K* queries. We do not need an associated index customized for each type of query; the proposed unified index is sufficient. In this section, we elaborate on how these queries are supported via our proposed index.

### 5.1. Spatiotemporal Range Query Processing

Spatiotemporal range query is the most basic query type to search for environmental data. This query type concerns locating IoT sensor data that are already recorded in a user-specified spatial region for a specified temporal interval. One of the most typical examples is, "Retrieve PM10 values measured in the (selected) area of Jung-gu, Daegu, South Korea, on November, 15th, 2017, from 6 to 9 pm." Currently, we allow a user to select such a spatial region in various forms: point, circle, line (road), trajectory, polygon, etc.

Algorithm 3 illustrates our spatiotemporal range query processing algorithm, implemented by the function *locateTrieNodes*(). This algorithm takes as input a user-selected spatial area ($S$), a specified temporal interval ($T_s$, $T_e$), the height of the index ($h$), and the root node ($R$). The algorithm first initializes the answer node set ($N_{ans}$) and obtains a list ($C$) of S2 cell IDs matching the selected area $S$ via the S2 Geometry libraries S2RegionCoverer, S2Loop, and S2CellUnion (Lines 1–2). Then, it pre-scans trie nodes with $C$ (Line 3). The reason for this is that S2 cells, where data do not exist, can be removed from the search scope to speed up the query. More details about the pre-scan are described in Algorithm 4. Next, for each STC ID $c$ in the pre-scanned set ($C_{ps}$), the algorithm populates the answer set $N_{ans}$ (Lines 5–13). To do so, we compute two respective index keys ($key_s$ and $key_e$) on the start and end times of the given interval (Lines 6–7). We then find the leaf node $N$, which is the starting point of the search, using $K_s$ (Line 8). Thereafter, we keep adding in $N_{ans}$ and updating the running node $N$ via its sibling pointer until its key is equal to $key_e$ (Lines 9–12). Lastly, we return the final answer set.

Asymptotic analysis: The time complexity of the algorithm is equal to $O(c \cdot M + P)$, where $c$, treated as a constant, is the length of bits used for an index key (STC ID); $M$ is the count of the S2 cells at Level 20 in a given spatial area $S$; and $P$ is the temporal range of ($T_s$, $T_e$). Here, note that $c$ can be further reduced; when the search starts, the prefix part of a selected area ($S$) can be easily skipped. The prefix means the common prefix of an S2 cell in $S$ (e.g., prefix 357 for South Korea and prefix 89$c$3 for New York in hexadecimals).

Algorithm 4 indicates the aforementioned pre-scanning method, called `getPrescannedSet()`. For each S2 cell ID $c$ in the input set $C$, we build a set ($N_{ps}$) of pre-scanned nodes (Lines 3–11). Specifically, in the for loop, the method performs a search with $c$ and obtains $N$ (Line 4). We then calculate the prefixes $P_N$ and $P_c$ for pre-scanning (Lines 5–6). We check if $P_N$ includes $P_c$ (Line 7). If so, we further check if the level of $c$ is less than 12. If it is, then a recursive call is made with the child of $N$ ($N.child$) (Line 8). Otherwise, we add the running node $N$ in $N_{ps}$ (Line 9). If $P_c$ is outside $P_N$, then we continue the for statement. For that reason, $P_N$ does not contain a common prefix of $P_c$, meaning that data do not exist inside the ST-Trie above the S2 cell $c$. We set the recursion base line to Level 12. This means that it is inefficient to make the recursion call on S2 cells at levels lower than 12. Finally, the built pre-scan set is returned (Line 12).

---

**Algorithm 3:** Spatiotemporal range query processing algorithm.

    **input** : A selected spatial area $S$, and a temporal range $(t_s, t_e)$, the index height $(h)$, and the
            index root node $(R)$
    **output**: The answer set of ST-trie nodes $(N_{ans})$

1 **Function** `locateTrieNodes`$(S, t_s, t_e, h, R)$:
2   $N_{ans} \leftarrow \{\}$ ;
3   $C \leftarrow$ `getS2CellIDSet`$(S)$;　　　　　　　　　// Obtain the matching S2 cells via library.
4   $C_{ps} \leftarrow$ `getPrescannedSet`$(C)$;　　　　　　　　　　　　// See Algorithm 4
5   **foreach** *STC ID $c \in C_{ps}$* **do**
6       $key_s \leftarrow$ `computeIdxKey`$(c, t_s)$;　　　　　　　// Compute an index key on $c_i$ and $t_s$.
7       $key_e \leftarrow$ `computeIdxKey`$(c, t_e)$;　　　　　　　// Compute an index key on $c_i$ and $t_e$.
8       $N \leftarrow$ `Search`$(key_s, h, R)$;　　　　　　　　　// See Algorithm 1
9       **while** *($N \neq NULL$) && ($N.key \leq key_e$)* **do**
10         $N_{ans}$.add$(N)$ ;
11         $N \leftarrow N.next$;　　　　　　　　　　　// Get the index key of $N$.
12       **end**
13   **end**
14   **return** $N_{ans}$ ;

---

**Algorithm 4:** Node pre-scan algorithm.

    **input** : A set of S2 Cell IDs $C$, the index height $(h)$, and the index root node $(R)$
    **output**: The pre-scanned node sets, $N_{ps}$

1 **Function** `getPrescannedSet`$(C, h, R)$:
2   $N_{ps} \leftarrow \{\}$ ;
3   **foreach** *S2 Cell ID $c \in C$* **do**
4       $N \leftarrow$ `Search`$(c, h, R)$ ;　　　　　　　　　　// See Algorithm 1.
5       $p_N \leftarrow$ `computePrefix`$(N.key)$;
6       $p_c \leftarrow$ `computePrefix`$(c)$;
7       **if** $p_c \subset p_N$ **then**
8         **if** *$N.level < 12$* **then** `getPrescannedSet`$(N.child)$ ;　　// Call recursively.
9         **else** $N_{ps}$.add$(N)$;
10       **else continue**;
11   **end**
12   **return** $N_{ps}$ ;

---

**Algorithm 5:** *k*-nearest neighbor query processing algorithm.

**input** : A spatial area $S$, a temporal range $(t_s, t_e)$, the number of nearest neighbors $k$, the index height ($h$), and the index root node ($R$)

**output**: The retrieved trie nodes $N_{ans}$

1   $N \leftarrow$ `locateTrieNodes`$(S, t_s, t_e, h, R)$;              `// See Algorithm 3`

2   $L \leftarrow$ Calculate the MBR center coordinates on $S$.

3   $N_{ans} \leftarrow \{\}, M \leftarrow \{\}$;          `// M: A map of elements of C and their distances` $d_i$

4   **foreach** *index node* $n_i \in N$ **do**

5      $d_i \leftarrow$ `computeDistance`$(n_i, L)$;     `// HaversineFormula is applied to compute distance.`

6      $M$.add$(n_i, d_i)$;

7   **end**

8   $M' \leftarrow$ Sort the entries of $M$ by their value $d_i$.

9   **foreach** *map entry* $(n_j, d_j) \in M'$ **do**

10     $N_{ans}$.add$(n_j)$ ;

11     $M$.delete$(n_j, d_j)$ ;

12     **if** $\left(|N_{ans}| == k \;||\; |M| == 0\right)$ **then** break ;      `// k neighbors found or no more entry left`

13   **end**

14   **return** $N_{ans}$ ;

---

## 5.2. k-Nearest Neighbor Query Processing

We support *k*-nearest neighbor (*k*-NN) query processing utilizing our proposed ST-Trie index. One typical example is, "Retrieve per-sensor PM2.5 values measured in the nearest *k* sensors to the (selected) area of Jung-gu, Daegu, South Korea, on November, 15th, 2017, from 6 to 9 pm."

Algorithm 5 describes the details. The algorithm begins by retrieving a set ($N$) of index nodes via the spatiotemporal range scan specified in Algorithm 3 (Line 1). We then calculate the centroid ($L$) of the MBR of the given spatial area ($S$) (Line 2). For each index node ($n_i$) in $N$, we compute the distance ($d_i$) between the spatial coordinates of $n_i$ and $L$ through the Haversine formula and record an entry of ($c_i$, $d_i$) into the result map ($M$) (Lines 4–6). After that, the map entries are sorted in ascending order of distance (Line 8). For each entry of ($n_j$, $d_j$) in the sorted map ($M'$), we add $n_j$ in $N_{ans}$ until (1) all the *k* neighbors are filled, or (2) there are no more entries left (Lines 9–13).

## 5.3. Top-k Query Processing

The last type of query is top-*k*, which retrieves IoT data in the ascending or descending order of their measured values. A typical example of top-*k* query is, "Retrieve the biggest $CO_2$ values measured in the (selected) area of Jung-gu, Daegu, South Korea, on November, 15th, 2017, from 6 to 9 pm."

Algorithm 6 details the top-*k* query processing. Basically, the algorithm takes the same input parameters as Algorithm 5 with one parameter representing a certain kind of thing ($p$) we collect from IoT sensors, for example PM2.5, PM10, temperature, humidity, $NO_2$, $CO_2$, and traffic counts. Like Algorithm 5, we start by searching for the nodes through the index (Line 1). We then extract the requested kind of thing data from the data records (at the bottom in Figure 3) pointed at by the located index nodes (Line 2). Finally, we complete by slicing the top-*k* thing data after the sort (Lines 3–4).

In the following section, we present the performance evaluation results of the described spatiotemporal range, k-NN, and top-*k* query processing via the proposed index.

---

**Algorithm 6:** Top-*K* query processing algorithm.

**input** : A spatial area *S*, a temporal range ($t_s$, $t_e$), the number of results to be shown *k*,
the name of thing *p*, the index height (*h*), and the index root node (*R*)

**output:** Top *k* records of *p*

1 $N \leftarrow$ `locateTrieNodes`(*S*, $t_s$, $t_e$, *h*, *R*);                    `// See Algorithm 3`

2 $T \leftarrow$ `extractThingData`(*N*, *p*);

3 $T' \leftarrow$ Sort *T* by *p*'s value.

4 **return** $T'[: k]$;                    `// Slice $T'$ up to $k$ from the top.`

---

## 6. Experiments

We first describe the experiment setups including the datasets used, the server machine specification, and the development language. Then, we evaluate the performance of the proposed index under various setups and discuss the evaluation results. We also analyze the indexing overhead and spatial efficiency when the index is compressed.

### 6.1. Environment Settings

Datasets: For our evaluation, we use three real-world spatiotemporal datasets. The first dataset, termed Daegu Taxi, concerns environment data collected while several taxis with attached IoT sensors moved through Daegu, South Korea. The second dataset, termed Porto Taxi, which is publicly available at the site [45], includes trajectory data obtained from taxis in Porto, Portugal. The last dataset, New York Taxi, includes trajectory data created by taxis in New York [46]. More details about these datasets can be found in Table 2. Every data record in all of these datasets includes a latitude, a longitude, and timestamp values. That said, some records were beyond the normal range of latitude and longitude values, so we removed them from the respective datasets. Moreover, the Daegu Taxi dataset includes various environment thing data, such as PM2.5, PM10, $CO_2$, $NO_2$, temperature, humidity, etc. We loaded these datasets into a single-node MongoDB 4.2.5 server and used them for evaluation.

**Table 2.** Real-world datasets.

| Attributes | Daegu Taxi | Porto Taxi [45] | New York Taxi [46] |
|---|---|---|---|
| # of Points | 40,077,744 | 83,409,386 | 1,363,569,980 |
| # of Records | 24,754,315 | 83,409,386 | 681,784,990 |
| Raw Size | 35.3 GB | 24.4 GB | 576.5 GB |
| Time Span | 2017/06/09–2019/09/29 | 2013/07/01–2014/06/30 | 2010/01/01–2013/12/31 |

Development language: The proposed ST-Trie index code was developed in C++. The C++ version C++14 was used to implement the trie index as a vector. The gcc compiler version was 7.5.0.

Server specification: All our experiments were performed on a workstation server, equipped with Intel Xeon Gold 5118 CPU 2.30 GHz, 256 GB RAM, and a 2 TB hard disk and running Ubuntu 18.04.2 LTS.

Testing environment: To evaluate the performance of our index, we chose two other competitors. One is the single-threaded HOT [37] and the other is the compound index of the 2dsphereindex for spatial data and the B-tree-based index for temporal data. In an effort to include more competitors, we cordially asked the authors of [24,35] to share their source code for comparison, but they did not respond to our favor.

For the evaluation, we performed each randomly generated query 50 times for fairness and took the median of the results [47] and used it as the query time. The parameters used in the queries are described in detail in the subsequent sections.

*6.2. Performance Evaluation*

This section shows the detailed evaluation results on the various types of spatiotemporal queries over the datasets used.

6.2.1. Spatiotemporal Range Queries

Figure 4 demonstrates the results of spatiotemporal range query processing under a variety of conditions. We compared the performance of our work, denoted as ST-Trie, with those of the two competitors, denoted HOT and 2dsphere + time, respectively. We expanded the spatial range of latitude and longitude used for comparison from $0.01°$ to $0.07°$. The range corresponds to 1 km$^2$, 16 km$^2$, and 49 km$^2$. For example, the spatial range $0.01° \times 0.01°$ represents an area with the latitude range $[40.72(5387), 40.73(5387)]$ and the longitude range $[-73.99(7359), -73.98(7359)]$. We also increased the temporal interval from one hour up to 10,000 h, or over a year, by a factor of 10. The 10,000 h interval corresponds to the maximum interval of the Porto Taxi dataset. Both the *x*- and *y*-axes are expressed in log scale.

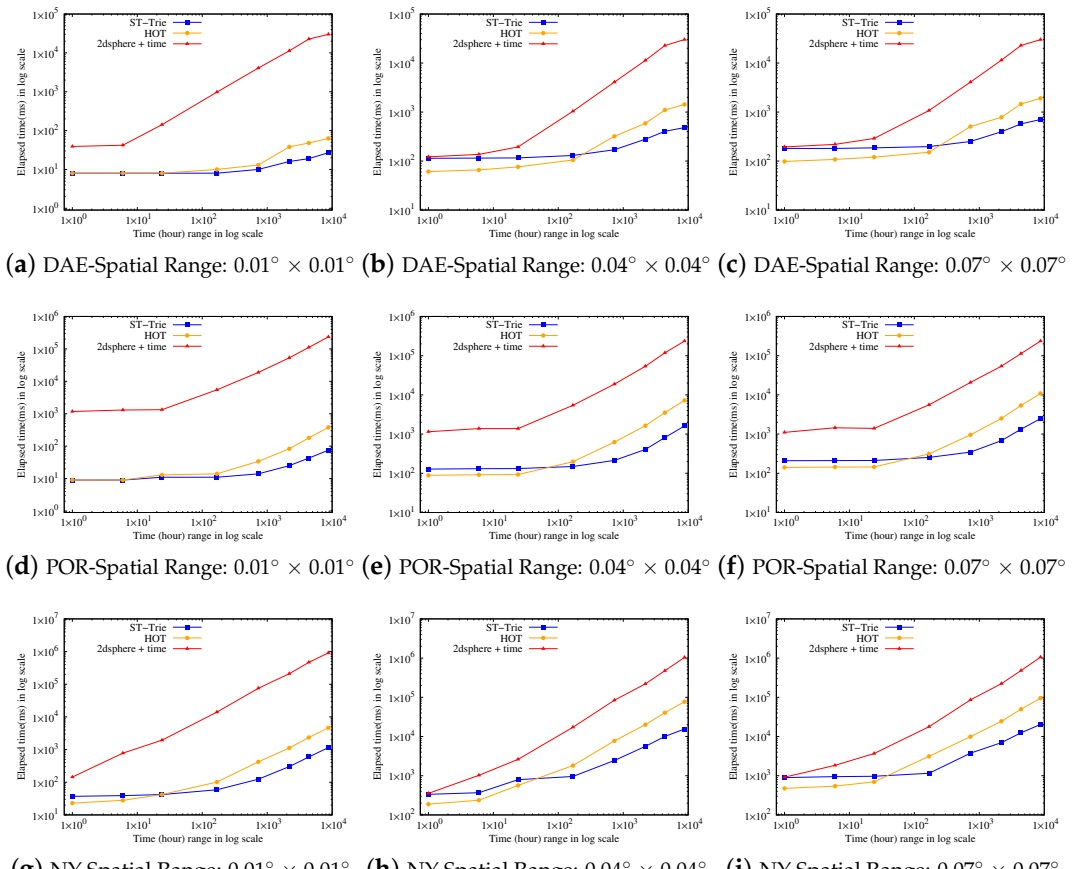

(**a**) DAE-Spatial Range: $0.01° \times 0.01°$ (**b**) DAE-Spatial Range: $0.04° \times 0.04°$ (**c**) DAE-Spatial Range: $0.07° \times 0.07°$

(**d**) POR-Spatial Range: $0.01° \times 0.01°$ (**e**) POR-Spatial Range: $0.04° \times 0.04°$ (**f**) POR-Spatial Range: $0.07° \times 0.07°$

(**g**) NY-Spatial Range: $0.01° \times 0.01°$ (**h**) NY-Spatial Range: $0.04° \times 0.04°$ (**i**) NY-Spatial Range: $0.07° \times 0.07°$

**Figure 4.** Query results over increasing spatiotemporal ranges on the various datasets: Daegu Taxi (DAE), Porto Taxi (POR), and New York Taxi (NY).

Figure 4a,c display the results on our dataset, Daegu Taxi. As the queried region in the spatial and temporal dimension was larger, our ST-Trie outperformed the two competitors. Specifically, ST-trie overwhelmed the composite index of 2dsphere + time by up to two orders of magnitude and bettered HOT by up to 2.7 times at the largest spatiotemporal region of $0.07° \times 0.07°$ and 10,000 h.

Nevertheless, ST-trie was slightly behind HOT below a certain point between the intervals of 100 h and 1000 h across the growing spatial windows. Note, however, that the *y*-axis is in log scale,

and the time resolution is in milliseconds. Therefore, the performance gap between HOT and ST-Trie is almost negligible.

Similar trends were also observed for the Porto and New York Taxi datasets, as shown in Figure 4d,i. In particular, for a spatiotemporal query with over one week on the New York Taxi dataset, a golden cross between the performance of HOT and ST-Trie occurred, and the margin steadily held until the end, as shown in Figure 4g,i. For Porto Taxi, our query processing via ST-Trie was 4.4 and 95 times faster than those of HOT and 2dsphere + time, respectively. Furthermore, ST-Trie presented better performance by about 4.7 and 51.7 times than HOT and 2dsphere + time for the New York Taxi data, respectively.

Query example: Figure 5 illustrates the results of processing a specified spatiotemporal range query on the Daegu Taxi dataset, using the proposed index. The spatial condition of the query was the circular area, somewhere in "Jung-gu, Daegu, South Korea", and the temporal interval of that query was "14 November 2017, from 6 pm to 6 pm the next day". In the figure, the blue, green, and yellow cells indicate low, intermediate, and severe microdust density, respectively. From these results, it is easy to notice which cells reveal more severe microdust density, compared to other cells in the circle.

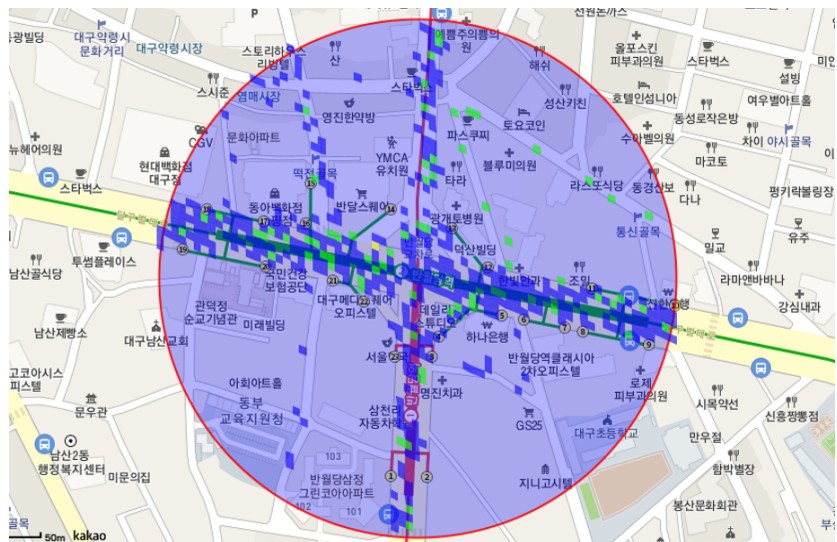

**Figure 5.** Spatiotemporal range query results on the Daegu Taxi dataset. Note that the building and road names on the above map [48] are in Korean (no English version of the map was available).

### 6.2.2. *k*-NN Queries

Figure 6 compares *k*-NN query processing performance between our proposed index and the compound index of 2dsphere and time. In the experiment, the spatial range was set to $0.01° \times 0.01°$ and the temporal range to one day. These parameters can be changed along with the user input. As can be seen in the figure, it took more time to answer a *k*-NN query over a bigger dataset. Even though the New York Taxi dataset has 27.5 times more records than the Daegu Taxi dataset, the time taken to serve a *k*-NN query was almost 17-fold. That is because for each *k*-NN query's processing, Algorithm 5 performs the spatiotemporal range scan via Algorithm 3.

The results indicate that, overall, ST-Trie overwhelmed the compound index on *k*-NN query processing performance. For instance, a *k*-NN query was served on our index about 800 times faster than the baseline compound index for the Daegu Taxi data. Although the margin was somewhat reduced, the performance gap still persisted for the other two datasets.

As *k* increased, both approaches needed slightly more time, but there was almost little difference in query processing.

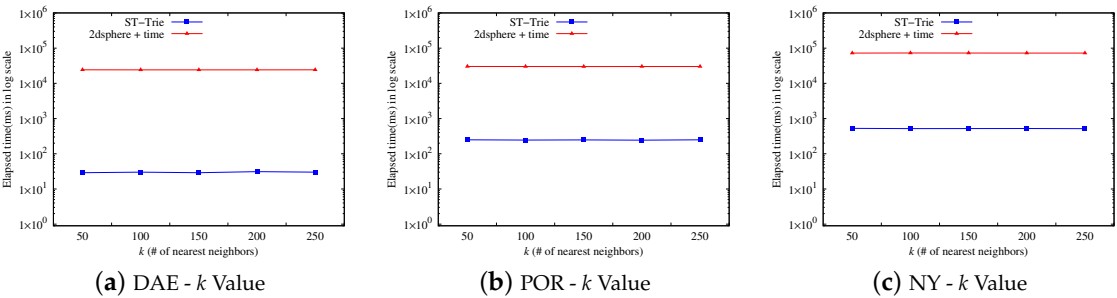

**Figure 6.** Performance of *k*-NN query on the various datasets: Daegu Taxi (DAE), Porto Taxi (POR), and New York Taxi (NY).

Query example: Figure 7 represents the cells and their associated microdust density retrieved by a specified *k*-NN query with the same spatiotemporal condition used in Figure 5 and *k* = 50. As illustrated in the figure, the 50 cells closest to the central point of that *k*-NN query were chosen, and their microdust values were extracted and represented in color. These results demonstrate that our *k*-NN query processing via the proposed index behaves correctly.

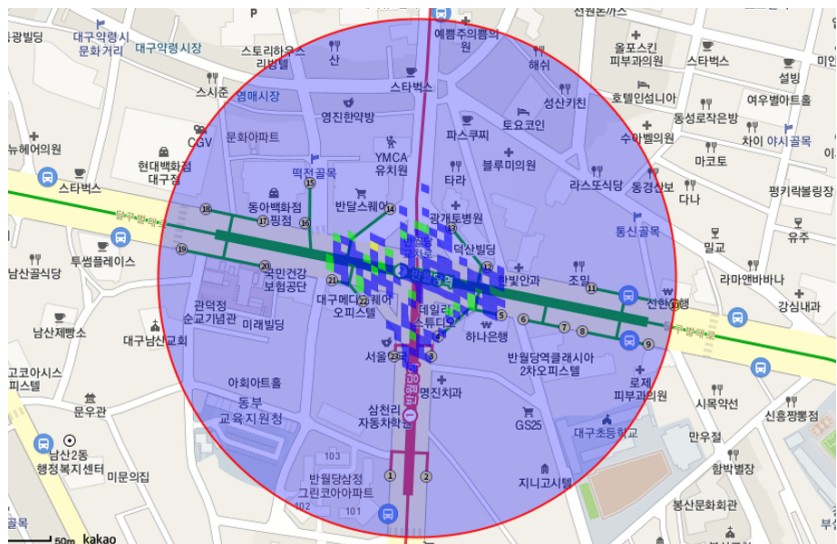

**Figure 7.** *k*-NN query results on Daegu Taxi.

### 6.2.3. Top-*K* Queries

Figure 8 illustrates the experiment results on top-*K* query processing through our proposed index. In this experiment, we incremented *k* from 50 to 250 by steps of 50 as configured in a recent paper [24]. The same spatiotemporal range parameter as specified in Section 6.2.2 was also set. As seen in the bar graphs, the overall query time increased with a bigger data size, as more spatiotemporal range query calls were made to locate the most *k* nearest records over the growing data. That said, the query performance was not affected by the varying *k*. Even though *k* increased by five times, the top-*k* query time was almost flat; different *k* values had little impact on elapsed time. From this outcome, we found that the proposed index kept the performance steady over a larger data size regardless of a greater *k*.

Query example: Figure 9 exhibits the cells and their associated microdust density chosen when a specified top-*k* query with the same spatiotemporal condition used in Figure 5 was issued with *k* = 50. As shown in the figure, the 50 cells with the highest microdust density in the selected area were chosen, and their microdust values are visualized with different colors. These results demonstrate that our top-*k* query processing leveraging the proposed index works very well.

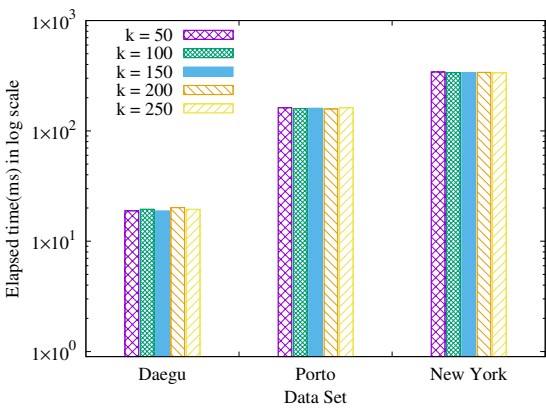

**Figure 8.** Top-*k* query performance.

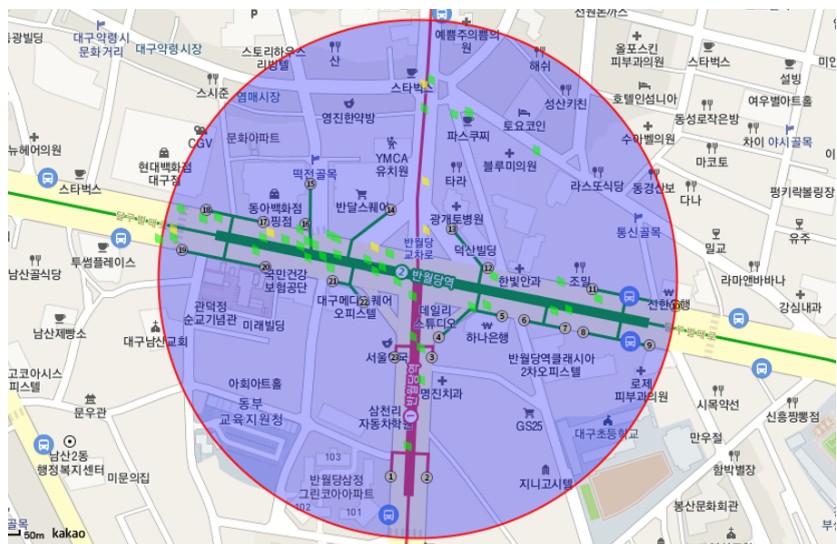

**Figure 9.** Top-*k* query results.

### 6.2.4. Indexing Overhead

We now study how much space overhead is incurred by our indexing scheme. For the growing datasets of Daegu Taxi, Porto Taxi, and New York Taxi, Figure 10 illustrates the memory and time overhead of the respective indexes. While varying the volume of each dataset from 20% to 100% by increments of 20%, we traced the memory usage and construction time of the respective indexes. As a result, overall, indexing time and memory size scaled very well with the growing data. More specifically, the index for Daegu Taxi with 20% of its data needed as much memory as 1 GB, and it took four seconds to build. The memory usage and construction time went up linearly as the data reached up to 100%. The indexing time and memory size for Porto Taxi also scaled very well. It took about seven seconds and 1.4 GB disk space to build an index over 20% of its data and 35 s and 6.5 GB disk space to index over 100% of its data. The indexing time scaled up linearly from 84 s to seven minutes, and the memory usage from 13 GB to 62 GB, for the New York Taxi dataset. Interestingly, the memory overhead was saturated as the full size drew closer. From these results, ST-Trie demonstrated solid scalability across the three different datasets as far as time and space overheads are concerned.

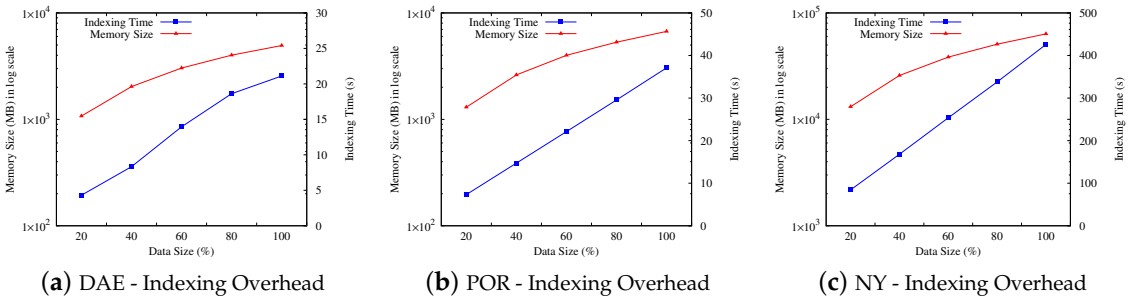

(**a**) DAE - Indexing Overhead     (**b**) POR - Indexing Overhead     (**c**) NY - Indexing Overhead

**Figure 10.** Indexing overhead on the various datasets: Daegu Taxi (DAE), Porto Taxi (POR), and New York Taxi (NY).

### 6.2.5. Compression Effect

Finally, we discuss the compression effectiveness of ST-Trie. The proposed index supports serialization by being permanently saved onto disk. Certainly, it is possible to restore the index from a disk through deserialization. Considering that (1) the physical structure of ST-Trie is composed of vector elements and (2) IoT data are typically massive, we wished to examine how the compression of the index could affect the storage consumption when it gets serialized.

Figure 11 depicts the compression efficiency of ST-Trie. As shown in the graphs, the compressed version (denoted by ST-Trie$_c$) of ST-Trie, on average, needed about seven times less storage than when uncompressed. Originally, the index size for Daegu Taxi was 5 GB, but when compressed, the size was reduced to 680 MB (7.35 times less). The compressed size was 920 MB (7.06 times less) and 8.5 GB (7.29 times less), whereas the uncompressed size was 6.5 GB and 62 GB for Porto Taxi and New York Taxi, respectively. The larger the data size was, the greater its compression efficiency in absolute size.

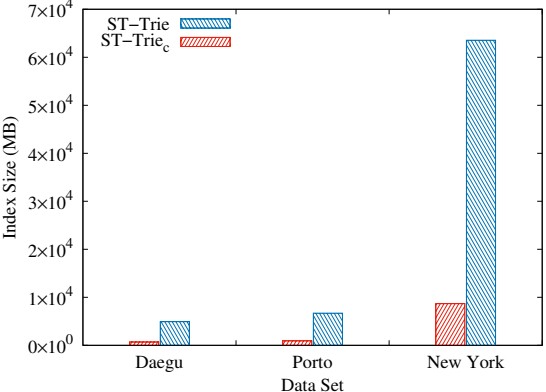

**Figure 11.** Compression Efficiency: "ST-Trie" indicates a uncompressed version, while "ST-Trie$_c$" means a compressed version.

## 7. Conclusions and Future Work

Recent IoT sensors have been continuously monitoring and generating not only various kinds of environmental data, but also traffic load data. Multi-dimensional analyses of such data can provide valuable insight into discovering new kinds of sustainable application services and enhancing our urban lives for better sustainability. Hence, it is of critical importance to support efficient retrieval over huge IoT data for easing these analyses.

To this end, in this article, we presented a novel indexing scheme, termed ST-Trie, over spatiotemporal IoT data. We introduced our innovative 3D-to-1D encoding method. In the method, the longitude and latitude values of given IoT data are converted to their corresponding S2 cell IDs, and then, the index keys are generated by keeping the upper bits of the obtained S2 IDs for spatial information and replacing the lower bits for temporal information of the original data. We then

presented a trie-based index structure built with the encoded keys, in which leaf nodes are connected to each other for fast time traversal. We also proposed the algorithms for processing various kinds of queries via the proposed index. In our experiment with three real-world spatiotemporal datasets, our scheme bettered the competitors in terms of query response time for the larger spatiotemporal regions. The proposed scheme also scaled very well, and the index compression turned out to be efficient. Lastly, the index overhead was acceptable as well.

In the future, we hope to support more types of queries, such as trajectory similarity and thing-based queries, via the proposed index. Furthermore, various datasets can help us better understand the performance of our index. Finally, we plan to further improve the performance when a narrower spatial region is specified and reduce the space overhead by removing the jump pointer of each node.

**Author Contributions:** Conceptualization, Y.-K.S. and R.L.; data curation, R.-Y.J. and M.P.; funding acquisition, Y.-K.S. and R.L.; investigation, H.C. and Y.-K.S.; methodology, H.C., J.K., S.K., and Y.-K.S.; project administration, Y.-K.S. and R.L.; resources, H.C.; software, H.C., J.K., and S.K.; supervision, Y.-K.S. and R.L.; writing, original draft, H.C. and Y.-K.S.; writing, review and editing, Y.-K.S. and R.L. All authors read and agreed to the published version of the manuscript.

**Funding:** This work was supported by an R&D project, "Enabling a System for Sharing and Disseminating Research Data (K-20-L01-C04-S01)" of the Korea Institute of Science and Technology (KISTI), Korea.

**Conflicts of Interest:** The authors declare no conflict of interest.

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
