# Peer review of "ST-Trie: A Novel Indexing Scheme for Efficiently Querying Heterogeneous, Spatiotemporal IoT Data"

_sustainability, doi:10.3390/su12229727_

Round 1
Reviewer 1 Report
The authors proposed a new indexing scheme for spatiotemporal data. It seems that the introduced indexing does improve query efficiency, time wise and space wise. The presentation of the research are smooth and sound. They also conducted adequate experiments to test their indexing scheme. According to the three datasets they have tried, their indexing scheme works well.
I feel the research methodology is sound, the results seems convincing to me. I am not a native speaker of English myself, but I feel that some sentences are somewhat awkwardly written, so language wise, the paper can be improved.
Reviewer 2 Report
Title: ST-Trie: A Novel Indexing Scheme for Efficiently Querying Heterogeneous, Spatiotemporal IoT Data
Authors present a novel indexing scheme (ST-Trie) for efficient retrieval over heterogeneous spatiotemporal IoT environment data. The advantages of the proposed approach are clearly described with respect to the existing indexing methods. An exhaustive evaluation of the proposed solution is presented achieving excellent results compared to the main state-of-the-art approaches.
Comments:
- The paper is very well-written, it is easy to follow and the organization is adequate.
- The motivation and the value added by the proposal are clearly described in the paper in “Introduction” section. The paper contributions are clearly described at the end of the previous mentioned section.
- The improvements with regard to related work are described.
- The proposal is very interesting and it is clearly explained.
- The evaluation is excellent.
I have just a few issues that may help to improve the paper:
- Section 1 Introduction. At the end of the section the contributions are well presented. Moreover, I think that a new contribution could be added (line 76). “We describe the Query Processing Algorithms for different kinds of queries”.
- Section 3 Approach. I think that the section title could be more specific. For instance: “Section 3. Our approach: the novel indexing scheme ST-Trie.”
In section 3.2 (line 217), Table 2 must be replaced by Figure 2; the example of a MongoDB document is not a Table.
- Section 4 Proposed Scheme. I think that the section title could be more specific. For instance: “Section 4. The proposed scheme: search and insert operations of ST-Trie.”
- Section 7 Conclusion and Future Work. It is necessary to relate the proposal to sustainability and sustainable development (the main aim of the journal). In Introduction section some ideas are presented but it is also necessary to emphasize them in Conclusion section.
- References must be reviewed and completed:
Add section title in line 513 “References”.
In websites, please, add the access date (lines 539-546).
- No comments in the remaining sections.
Minor spell check. English must be reviewed by native speakers.
- Line 223. The sentence must be reviewed. “That said, we should mention that our encoding scheme depends on such a specific NoSQL storage nor a relational database.”
- Line 394. “…respectively. we expanded…”. Capital letter omitted.
